# Overview of Wearable Healthcare Devices for Clinical Decision Support in the Prehospital Setting

**DOI:** 10.3390/s24248204

**Published:** 2024-12-22

**Authors:** Rachel Gathright, Isiah Mejia, Jose M. Gonzalez, Sofia I. Hernandez Torres, David Berard, Eric J. Snider

**Affiliations:** Organ Support and Automation Technologies Group, U.S. Army Institute of Surgical Research, JBSA Fort Sam Houston, San Antonio, TX 78234, USA

**Keywords:** wearable healthcare device, medical monitoring, medical decision support, prehospital, machine learning, advanced monitoring, physiological signals

## Abstract

Prehospital medical care is a major challenge for both civilian and military situations as resources are limited, yet critical triage and treatment decisions must be rapidly made. Prehospital medicine is further complicated during mass casualty situations or remote applications that require more extensive medical treatments to be monitored. It is anticipated on the future battlefield where air superiority will be contested that prolonged field care will extend to as much 72 h in a prehospital environment. Traditional medical monitoring is not practical in these situations and, as such, wearable sensor technology may help support prehospital medicine. However, sensors alone are not sufficient in the prehospital setting where limited personnel without specialized medical training must make critical decisions based on physiological signals. Machine learning-based clinical decision support systems can instead be utilized to interpret these signals for diagnosing injuries, making triage decisions, or driving treatments. Here, we summarize the challenges of the prehospital medical setting and review wearable sensor technology suitability for this environment, including their use with medical decision support triage or treatment guidance options. Further, we discuss recommendations for wearable healthcare device development and medical decision support technology to better support the prehospital medical setting. With further design improvement and integration with decision support tools, wearable healthcare devices have the potential to simplify and improve medical care in the challenging prehospital environment.

## 1. Introduction

Medical diagnosis and treatment options are based on available data when assessing a patient. These decisions can be faulty if the information collected from a subject is incomplete or inaccurate. Patient vital signs are a critical component in the medical decision process and are used to make well-informed medical decisions [1]. However, in the prehospital environment where trauma must quickly be classified so that life-saving interventions and treatments can be administered, information on vital signs is less available [2]. This is true for prehospital services in the civilian sector, but even more so during combat casualty care. Military medicine requires a high level of care to be provided outside of a traditional hospital environment with limited medical resources [3,4]. With the ongoing Ukraine–Russia conflict, prolonged casualty care in theater has been drawn out due to challenged air space and long-range artillery making patient transport difficult [5].

Prehospital challenges related to vital sign monitoring are further compounded by the complexity in medical decision-making during trauma cases. For example, during early stages of trauma-induced hemorrhage, physiological parameters can be influenced by fear, pain, or other unknown factors. Compensatory mechanisms are highly subject dependent and can complicate early triaging by disguising shock symptoms until life-saving interventions may no longer be effective [6]. These physiological compensation mechanisms function by protecting oxygen delivery to vital organs at the expense of peripheral requirements, via sympathetic vasoconstriction and increased extraction of oxygen from blood [7]. As such, vital sign trends are often more important than single readings so that changes can be more easily tracked. Therefore, technologies that allow earlier, more frequent, accurate, and objective measurements of physiological parameters are necessary to adequately triage hemorrhagic shock at early stages.

Towards this, wearable healthcare devices (WHDs) have the potential to greatly improve patient monitoring in the prehospital setting. These devices can easily be worn by a patient without hindering mobility and are often wireless with small footprints and low power requirements [8,9]. Many WHDs have been commercialized or are in development for a wide range of physiological metrics. However, WHDs will need to be supplemented with advanced clinical decision support (CDS) tools built with machine learning (ML), artificial intelligence (AI) model architectures to translate complicated physiological patterns into more easily interpretable data [10].

WHDs have been the focus of a number of review articles detailing the wide variety of WHDs in development [9,11,12,13,14]. However, the integration of WHDs with prehospital civilian and military care and CDS algorithms has not been the focus of other review articles. One review article focused on WHDs and CDS applications but only as they pertained to COVID-19 prevention [15], while another recent review evaluated WHDs for paramedics but specifically for improved measurement of paramedic fatigue and not casualty care [16]. The scope for this review article is WHDs and their integration with AI/ML models for aiding with clinical decision support in the civilian and military prehospital setting. To meet this objective, we have structured this article into the following sections:An overview of prehospital medical care for both civilian and military situations.A detailed overview of clinical decision support tools.Method description on how WHDs and CDS applications were identified for this review article.Description of the most relevant WHD vital sign monitors including what the sensor(s) measures, a summary of state-of-the-art WHDs for each monitor type, and how the WHD has been used for clinical decision support.Discussion section highlighting where the current shortcomings exist for WHDs and CDS tools.

## 2. Overview of the Prehospital Medical Setting

Prehospital medical care differs from emergency medicine or other medical services primarily due to limited availability of resources. Ambulatory care or wilderness medicine has a high focus on portability of medical monitors [17]. Traditional medical monitoring systems do not meet this criterion and therefore cannot be readily used in these settings, suggesting a potential use for WHDs. Furthermore, few personnel may need to manage multiple trauma patients, leading to the importance of triage capabilities. The limited medical sensors available for civilian prehospital medicine creates challenges when prioritizing resources or identifying treatment options [18,19]. Integration of machine learning interpretation of sensor inputs could revolutionize prehospital medicine.

For combat casualty care, recent military conflicts, such as Operation Enduring Freedom, emphasized the need for timely medical interventions, specifically during the “golden hour” [20]. This critical first hour of medical intervention was the goal of the prehospital setting as nearly all casualties could be evacuated to a hospital within a few hours. However, the future battlefield against near-peer adversaries will have conditions where air space is contested, resulting in fewer opportunities to evacuate casualties from the combat zone [3,21]. Near-peer adversaries will result in multi-domain operations connecting each fighting domain—land, air, cyber, sea—into a single conflict, complicating military medicine greatly. In these settings, it is anticipated that prolonged field care of up to 72 h will become the standard. As such, improved patient monitoring, surgical techniques, and medical triage will be required in theater during prehospital combat casualty care.

The ongoing Ukraine–Russia conflict also exemplifies these challenges projected on the future battlefield. Limited medical evacuation opportunities have arisen due to challenged airspace, which requires far forward surgical teams to treat and manage a larger number of casualties for up to 72 h in theater [5]. This has been further complicated by precise, long-range weaponry minimizing the relative safety of casualty evacuation even at distances above 500 km away from enemy lines. In addition, more than 70% of Ukraine casualties stem from more advanced rocket or artillery injuries, which often result in complex polytrauma to multiple organ systems [5]. Thus, as we look towards the future battlefield, it is even more imperative that accurate triage procedures are used to improve physiological monitoring and prioritize injured warfighters during limited evacuation opportunities. For these reasons, WHDs and CDS tools will be critical for future prehospital care for both civilian and military situations.

## 3. Overview of Machine Learning for Clinical Decision Support

Patient monitoring can provide an incomplete picture of a casualty’s condition if vital sign trends over minutes, hours, or days are not interpreted by trained medical providers, who are less readily available in the prehospital setting. To address this need, machine learning (ML) models can be developed to provide advanced interpretation outputs based on vital sign inputs. These algorithms can be developed to predict easy-to-interpret metrics that could further support medical decisions. However, developing ML decision support models requires large clinical datasets that may not be easily attained [22]. Large, diverse datasets are critical for ensuring that models are generalized to a large patient population as needed for creating deployable CDS triage systems [23].

Different methodologies, from feature extraction to wavelet transformations, can be used as inputs to ML models to obtain the optimal medical decision response [24,25]. The use of ML to interpret these different inputs can not only improve performance but also provide additional physiological information not readily apparent with traditional medical information [26]. This, otherwise hidden, information extracted by the ML algorithms can aid a healthcare provider in supporting medical decisions. In addition to different input setups, ML model architectures can range from simple regression model setups to complex support vector machines or AI deep learning architectures with convolutional or recurrent neural network layers. ML models for signal processing have been the focus of a number of review articles and will not be discussed in detail here [27,28,29,30]. Instead, how ML models can be used to develop CDS tools will be highlighted.

CDS systems have been developed for a wide range of applications using ML models to interpret vital signals informing medical diagnostics and treatments. One common CDS application utilizing ML models is medical image diagnostic interpretation. Tools have been developed for providing decision support for tumor malignancy determination [31,32,33], foreign body proximity to neurovascular features [34], and COVID-19 diagnostics [35,36,37]. However, these examples all rely on large, bulky medical imaging equipment not currently available in the prehospital setting. Other CDS examples are related to vital sign signal monitoring for quantifying underlying compensatory effects on patient status to provide earlier identification of shock [38,39]. These algorithms relied on photoplethysmography or arterial blood pressure waveforms to predict compensatory status using deep learning or classical ML models [39,40,41]. An additional CDS tool for predicting hypotension was commercialized by Edwards Lifesciences using invasive arterial waveform data or non-invasive measurement analogues [42,43,44]. Another common CDS application is fluid management tools, wherein provided medical information and continuous vital signs can drive fluid resuscitation decisions. Burn resuscitation, while normally an intensive process requiring manual titration of fluids prone to error, has been simplified with a CDS tool to drive fluid infusion rates based on urine output rates [45,46]. Similar approaches have been used to develop CDS tools for anesthetic infusion [47,48], vasopressor infusion [49,50,51], and hemorrhage resuscitation [52,53,54]. While significant work has been performed related to CDS tools to improve medical care, many of these are not deployable in the prehospital setting without integration with portable WHDs first.

## 4. Methods

To meet the objectives of this research study, the authors and research team utilized a number of search approaches to identify WHDs that meet the scope of this paper. These approaches included PubMed and Google Scholar for identifying research studies on WHDs in development, as well as traditional search engines for identifying commercialized WHDs. For the purpose of this review, WHDs are defined as any non-invasive portable device that can stream data wirelessly and still allow the subject freedom of movement, excluding devices requiring continuous user interaction. Biosensors using microneedles or similar technology were disregarded as they were not considered to be non-invasive. The initial review of the type of WHDs and any potential prehospital application was reviewed by the research team to down-select to seven specific sensor technologies detailed in the following section.

Searches were then conducted to evaluate WHDs already developed or in development for these technology types. Focus was directed towards commercialized products as the scope of this review was for possible uses in the prehospital environment; as such, technologies already commercialized have a possibility of meeting this goal quicker. Similarly, studies discussing FDA clearance or validation testing were also of interest to identify which WHDs were closer to being used in prehospital medical applications. Results from these searches for each technology were reviewed by the authors to identify the most relevant technology for prehospital care. In addition, CDS applications were evaluated for each sensor technology. This took three different search phases. First, identification of CDS applications for WHDs with the sensor technology. Second, evaluation of any relevant CDS applications related to hemorrhage detection or other prehospital injuries for each sensor technology with a non-WHD format. Lastly, if no examples were found from the first two, CDS applications less relevant to prehospital applications were evaluated to understand the state of development and how they could transition to a prehospital setting. Results were compiled for each technology in the subsequent sections.

## 5. Wearable Healthcare Devices for Clinical Decision Support Tools

Wearable healthcare devices can be applied to different parts of the body as depicted in Figure 1 for sensing a wide range of vital signs. Major types of vital sign measurements that use WHDs, as relevant for prehospital medicine, are detailed in subsequent sections. Each section below will (i) provide a summary of the physiological measurement technology, (ii) describe CDS applications for the sensing mechanism, (iii) overview of state-of-the-art WHDs and their commercial availability, and (iv) discuss the current strengths and weaknesses of the WHD for the prehospital setting.

### 5.1. Electroencephalogram

#### 5.1.1. Overview of Technology

Electroencephalograms (EEGs) are used for monitoring or identifying abnormal brainwaves via detection of electrical activity in the brain. Electrodes are placed on the scalp for monitoring electrical charges associated with normal brain function or evoked potentials, i.e., neural activity triggered by sensory stimuli. The signals are processed and appear as an interpretable graph which can help identify abnormal trends or acutely deficient responses to stimulus [55]. EEG readings have been shown to associate with visual [56,57], auditory [58,59], and large-scale neuronal functional maladies, such as those resulting from traumatic brain injury (TBI), if the data are properly processed [60,61,62].

When compared to other brain diagnostic techniques, such as magnetic resonance imaging (MRI) or computed tomography (CT) scans (see Section 5.3), EEGs have excelled as a non-invasive method for monitoring brain activity, particularly with regard to portability and cost benefits. EEG devices are traditionally composed of a cap containing metal electrodes that are dispersed across key locations that make contact with the scalp [63]. However, these caps usually require a physical connection to an external power supply and specialized hardware to process and record the data. Recently, the increased healthcare focus on portability and small form factor devices has prompted demand for an easy-to-use and even more compact EEG capability as a WHD.

#### 5.1.2. Medical Decision Support Applications

Monitoring patient conditions in a conventional hospital setting is costly and requires large, cumbersome equipment but can offer expansive data for future CDS model development [64]. Traditional EEG signals have been used with ML models for diagnosing conditions such as Alzheimer’s [65], epilepsy [66], and TBI [67]. Pairing these CDS tools with WHDs can provide early disease diagnosis or other health insights to inform triage in the prehospital setting. For instance, hemorrhage resuscitation, of critical importance during combat casualty care, would be impacted by diagnosis of a TBI as recommended target blood pressure for resuscitation must be adjusted in these situations [68,69].

One application of EEG WHDs using ML/AI models has been for seizure monitoring across multiple days using the Epitel Epilog system, which achieved 100% sensitivity and 70% specificity with the CDS algorithm [70]. Other EEG-based CDS system examples include diagnosis of brain diseases [71], assessing cognitive function [72], stress management, and obtaining neurological feedback [73].

#### 5.1.3. State of Development

The ease of use of wearable EEG technology has led to these WHDs being used as lifestyle devices, providing users with easy-to-understand metrics such as stress indications, while incentivizing individuals to adopt a healthier lifestyle. Development of WHDs for EEG sensing has advanced in recent years to small, portable commercially available devices. Traditionally, WHDs designed to implement EEG technology have a headband-style form factor with some using a headset containing additional electrodes on the upper scalp [74]. Portability has greatly improved as these WHD systems use wireless data transmission and built-in batteries, eliminating the extensive wiring that formerly hampered movement. Another advancement of EEG WHDs has been the use of dry electrodes in place of traditional wet electrodes, eliminating the need for conductive gel. This has greatly improved the ease of application for untrained personnel and enabled long-term monitoring that was previously limited by the gel drying out or rubbing off [75]. Wearable EEGs such as the BrainBit smart EEG headband (BrainBit Inc., New York, NY, USA), Muse EEG headband (Muse, Toronto, ON, Canada), Epitel’s Epilog (Epitel, Salt Lake City, UT, USA), and BitBrain Diadem (Bitbrain Inc., Zaragoza, Spain) are examples of devices currently on the market. All of these are relatively simple to set up, compact in size, and utilize a self-contained design which drastically expands the range of motion and potential usage locations compared to traditional EEG setups. However, they vary in data quality based on the quantity of measurable EEG channels which can limit their diagnostic reliability.

#### 5.1.4. Technology Advantages and Disadvantages

Early recognition of TBI is critical to ensure time-sensitive care is provided, and prehospital deployment of EEG has shown promise in delivering key information during triaging [76]. The clear advantage a WHD EEG system brings is making long-term continuous EEG monitoring available in far more remote settings than within a traditional hospital. Although, there is still a capability gap in expanding clinical-grade CDS tools developed for EEG systems to WHDs, particularly for TBI. Limitations of wearable EEG technologies include ensuring adequate and reliable electrode contact between the headband and the scalp which can be difficult with dry electrode setups due to the variability in patient skull sizes/shapes and interference from hair or skin conditions. Further, most available EEG WHDs use fewer electrode contacts than the international standard of 10–20 electrodes, reducing the sites where brain activity is monitored which hinders diagnostics that depend on targeted regions or correlating signals from multiple regions. As a result of fewer channels, restricted computational power, and limited filter capabilities, issues can present themselves. This can manifest as signal artifacts that make analyzing the data much more difficult. Applying highly capable ML/AI models can potentially address these challenges, but it will require additional training data collection and model development. A summary of EEG WHDs for the prehospital environment and CDS applications is shown in Figure 2.

### 5.2. Photoplethysmography

#### 5.2.1. Overview of Technology

Photoplethysmography (PPG) has the capability to non-invasively monitor blood volume changes in the microvascular bed of the tissue from several places on the body including ears, fingers, and cheeks [77]. Conventionally, PPG is measured using an infrared light source that is absorbed by blood and a photodetector to capture changes in light intensity. If the light source transmits through tissue and is captured via the photodetector on the opposite side, the PPG sensor is configured for transmission mode wherein only light sources that pass through the tissue are detected. PPG measurement is also possible in reflectance mode if the light source and detection are on the same side, adjacent to each other, allowing for application to a wider variety of profound tissues as the light does not have to transmit through the thickness of the tissue. The resultant PPG waveform captured by the photodetector has been well characterized to measure multiple physiological features. The pulsatile properties or alternating current (AC) components of the PPG signal correspond to changes in arterial blood flow as a result of cardiac activity while larger signal shifts, referred to as the direct current (DC) component, are driven by respiration, sympathetic nervous system activity, overall blood volume, and thermoregulation [78].

The PPG waveform includes a variety of features such as, but not limited to, systolic amplitude, peak to peak intervals, pulse area, and heart rate variability, which can all be extracted and used to predict vascular changes in the bloodstream [79]. Heart rate variability (HRV) can be obtained from a PPG signal and is used to determine the state of the patient from the effects of the autonomic nervous system. Factors that may affect HRV include cancer, age, and thermoregulation. Of particular interest for prehospital medicine, HRV has been shown to increase in the early stages of hemorrhage [80] and has been shown to be an early indicator of impeding septic shock [81]. The second derivative of the PPG signal, known as the acceleration photoplethysmogram (APG), indicates the acceleration of the blood volume changes or instantaneous power of the blood circulation [82]. The APG provides information on a patient’s arterial stiffness and other possible cardiovascular diseases [78]. These sensors can also provide non-invasive measurement of arterial oxygen saturation (SpO_2_) percentage by passing two distinct light wavelengths through the tissue, in the transmission configuration [83]. In critical care scenarios SpO_2_ is an essential parameter to monitor, and PPG sensors can provide continuous output of this measurement.

#### 5.2.2. Medical Decision Support Applications

PPG sensor inputs have been utilized for a wide variety of CDS applications as a standalone sensor or in conjunction with other vital sign monitoring. Most PPG-based CDS functions require extraction of waveform features such as the systolic peak and dicrotic notch rather than using the raw waveform, but these processes can be automated with ML models for real-time analysis [84]. However, with more robust deep learning architectures, more recent approaches have simplified the use of PPG signals by eliminating the need for feature extraction and utilizing the raw waveform [85]. Instances of CDS using a PPG input are vast including improved diagnosis of preeclampsia [86], prediction of sleep state for classifying sleep apnea severity [87], and assessing the degree of vessel stenosis [88]. For the prehospital environment, medical decision support tools have been developed for prediction of impending ventricular arrythmias minutes before their occurrence, allowing additional time for emergency intervention [89]. Other emergency applications include prediction of hypoxemia severity for aiding in COVID-19 assessment [90] or predicting hemodynamic decompensation onset for earlier indication of hemorrhagic shock [91,92]. Another avenue for developing novel CDS tools using PPG signal is the prediction of blood pressure and even reconstruction of the arterial blood pressure waveform, something only possible with invasive catheter placement or bulky monitoring equipment. These techniques have been extensively detailed in other review articles [93,94]. Any of the above techniques are likely suitable for use in wearable PPG sensor formats depending on the fidelity of data capture.

#### 5.2.3. State of Development

PPG sensors serve as non-invasive, inexpensive, and convenient diagnostic tools for cardiovascular health assessment. While commonly used finger or ear clip sensors are often used for PPG measurement, these systems cannot be considered wearable technology as they often tie into patient monitoring systems or are placed over the end of the finger, impeding daily functions. However, there are several PPG WHDs on the market or in development [78]. The most common form factor for PPG WHDs is smartwatches with countless manufacturers implementing PPG sensors for tracking SpO_2_ or heart rate. Notably, the Empatica EmbracePlus smart watch platform has been FDA-approved for patient monitoring [95]. Recent smart ring designs incorporate PPG signal measurement for SpO_2_ measurement with a reduced form factor. Other PPG WHDs include stickers with embedded light sources, photodetectors, and Bluetooth transmitters for wireless tracking of the SpO_2_ signal [96,97]. A wider variety of placement options arises from a sticker-based sensor compared to the smartwatch or ear clip formats. Technology is in development for magnetic earring PPG WHDs [98] as well as earbud-based sensors [99], including the Philips Sleep Headphones which can track sleep status via embedded PPG sensors [100].

#### 5.2.4. Technology Advantages and Disadvantages

PPG measurement can provide a waveform analogous in many applications to the arterial waveform as well as measurement of tissue oxygenation, both of which are critical for patient monitoring during emergency medicine applications. One key advantage of PPG technology is the wide range of ML tools already developed for CDS applications. PPG can be used as an input for early prediction of hemorrhagic shock or ventricular arrythmias. Another advantage for PPG sensing technology is the wide range of measurement locations available [101,102,103], which can be critical following certain injuries to allow for diagnostic interventions. PPG WHD placement flexibility, especially using sticker formats, can make their adaptation in prehospital medical applications more convenient. The primary limitation with PPG sensors is their high susceptibility to motion artifacts, which is particularly challenging during patient transport [104]. This results in the potential for inaccurate readings when tracking the PPG signal during sudden or unexpected movements. In addition, the PPG signal collection is susceptible to being negatively affected by environmental factors, such as electromagnetic interference, resulting in noise degrading the PPG signal [104]. However, there are extensive research efforts underway to filter out motion and other signal artifacts to overcome these challenges [104,105]. A summary of PPG WHDs for the prehospital environment and CDS applications is shown in Figure 3.

### 5.3. Medical Imaging

#### 5.3.1. Overview of Technology

Medical imaging is used in multiple aspects of patient and casualty care, including triage and treatment guidance, as it allows medical professionals to visually examine internal structures. There are multiple imaging technologies that are commonly used in emergency medicine settings, including ultrasound (US), X-ray, CT scans, and MRI, among others. Consistent and reliable access to any imaging modality is critical in a battlefield setting for monitoring worsening internal injuries and prioritizing evacuation decisions. Unfortunately, most medical imaging capabilities, with the exception of small portable US systems, cannot be practically utilized in prehospital settings due to their power requirements, cost, and large footprint [106]. US imaging uses a probe or transducer to emit high-frequency soundwaves into a tissue and measures the signal strength and time delay of the reflected soundwaves returning to the transducer [107]. As different tissues have different US echogenic properties [108], US imaging can be used for visualizing internal anatomical structures, measuring blood flow in central vessels, or measuring tissue stiffness [109,110]. Advanced ultrasound imaging techniques, such as contrast-enhanced ultrasonography and color ultrasonography, extend the capabilities of US by enabling the quantification of tissue perfusion and overall organ function. These are critical metrics for assessing the physiological state of patients in emergency and critical care settings [111,112]. While portable US systems may be accessible in the prehospital setting, they require skilled medical personnel for image capture and interpretation, resulting in intermittent information at best, as opposed to the continuous monitoring and interpretation that is possible with other sensing modalities. The development of medical imaging technologies in a wearable format, with computer-aided analysis or CDS, is essential for improving image-based triage in the prehospital environment.

#### 5.3.2. Medical Decision Support Applications

Medical imaging is traditionally used to aid in diagnosis by allowing the user to visualize any abnormalities within a patient’s body, such as a broken or fractured bone identifiable by X-ray, MRI, or CT scan during emergency medicine. Furthermore, medical imaging is a current standard of practice used to monitor disease progression and determine the efficacy of treatment. Most notably, if a patient has been diagnosed with cancer, medical imaging can be used to see its progression and determine how well the cancer cells respond to chemotherapy. However, these current methods rely on extensively trained and highly skilled professionals to acquire and interpret the US scans to infer conclusions about the state of the disease. As such, AI/ML-based CDS tools have been widely developed for medical applications and have been thoroughly reviewed elsewhere [113,114,115,116]. Some relevant CDS applications for prehospital medicine include automated US-based detection of abdominal hemorrhage [117,118], pneumothorax [119,120], and COVID-19 [36,37]. By redesigning these imaging devices to be in a wearable format, data can be continuously collected “hands-free”, opening the possibility to a wider range of CDS applications as the imaging WHD technology becomes more readily available.

#### 5.3.3. State of Development

Because traditional US techniques are already non-invasive and smaller, compared to other imaging modalities, there has been more success integrating this technology into wearable devices. However, challenges remain regarding the miniaturization of US technology while maintaining sufficient power for these devices. Initially, US WHDs involved physically mounting a full-sized US probe onto a person’s skin via a robot or strap setup; however, these approaches were shown to interfere with patient mobility and were both inconvenient and uncomfortable [121]. To address this, research and development of wearable US sensors has focused on elastic imaging devices in the form of skin patches or adhesive hydrogels. These devices comprise a very small transducer attached to the target area to provide real-time continuous monitoring, allowing for quicker intervention and personalized patient care [122].

One of the only commercially available wearable imaging devices on the market is the FloPatch™ (Flosonics Medical, Greater Sudbury, ON, Canada), a wearable Doppler ultrasound device that provides the user with real-time data regarding their hemodynamic metrics [123,124]. In a proof-of-principle experiment, the FloPatch US WHD measured Doppler spectra among patients who had undergone double and triple coronary bypass surgeries, with the intent of defining a relationship between the velocity time integral (VTI) and stroke volume [81]. Specifically, the device was able to accurately measure metrics such as VTI and carotid flow time for insight into sudden physiological changes due to trauma or illness. [81]. This is especially important because clinical applications have found usefulness in acute medicine where continuous hemodynamic recordings enable the user to measure changes in stroke volume in real-time.

In addition, a number of promising US WHD technologies are currently in development. One example is a device by the University of California San Diego that has a similar sticker wearable form factor and was able to measure blood pressure and cardiac output for up to 12 h continuously, even while the subject was moving [125]. Further, their technology was capable of imaging to depths of 16 cm, allowing for continuous imaging of deep tissue or organs such as the heart. Another device, still in development, is an ultrasound wearable comprising a thin rigid ultrasound probe attached to a tough bioadhesive hydrogel elastomer couplant. The design aims to allow for stable imaging resulting in better resolution and quality during body motion [121]. It can provide up to 48 h of continuous monitoring for various organs, including blood vessels, tissues, and organs [84]. It is comfortable to wear over prolonged periods of time and provides high-resolution images, making it a valuable diagnostic and monitoring tool for various diseases and developing injuries in both civilian and military settings.

In contrast to the adhesive patches that are more prevalent for WHDs, a different form factor in development is a neckband with US sensors placed at the carotid artery for wireless Doppler blood flow measurement [126]. Initial testing determined that the wireless neckband was comparable to traditional US machines regarding visual assessment of images and measured parameters. Moreover, it enabled continuous monitoring of blood flow dynamics in common carotid arteries and facilitated the monitoring of symptomatic or asymptomatic patients with cerebrovascular or cardiovascular diseases.

Each of these are in early stages of development but have the potential to revolutionize the medical field by gathering continuous, real-time data on the internal physiology of the user and offering overall health insights.

#### 5.3.4. Technology Advantages and Disadvantages

Wearable imaging sensors can provide many benefits in prehospital care, including continuous imaging of internal organs over longer periods of time to gain valuable insights into developmental biology and diseases. However, more research still needs to be performed before these devices can be reliably used in either military or civilian prehospital settings. Current wearable US devices are limited by low imaging resolution, unstable imaging quality due to artifacts such as those caused by patient movement, and limited battery life [121]. These shortcomings lead to inconsistent images with high signal noise, making it more difficult to diagnose a patient or infer conclusions regarding the user’s health. Advanced signal processing or ML approaches can be incorporated into WHD imaging devices to help isolate signal from noise. Unfortunately, gathering the large image datasets needed to develop these models can be logistically challenging and resource intensive. Additionally, the small-footprint, “stretchable” nature of some of the WHDs in development produces issues with their piezoelectric elements, leading to images with less stability and reduced resolution [121]. Lastly, the current method of adhering WHD US systems to a patient’s body involves a hydrogel or elastomer coupling which can easily become dehydrated, resulting in detachment in only a few hours [121]. By addressing these issues, US WHDs can revolutionize the healthcare industry by providing end-users and physicians with information on internal organ states and their dynamics to draw better conclusions about the patient’s health in both civilian and military medicine. A summary of US WHDs for the prehospital environment and CDS applications is shown in Figure 4.

### 5.4. Chemical Sensors

#### 5.4.1. Overview of Technology

Chemical sensors are used to detect and translate traditional protein or laboratory test results into measurable signals to obtain continuous recording of a signal of interest. Chemical sensing can be used in a wide array of applications, such as medical diagnostics, monitoring chronic conditions, in sports medicine to track electrolyte balance and dehydration, and in clinical settings to monitor patient metabolism and overall organ function [127]. Traditional modalities for measurement in these applications often involve invasive measurement methods and laboratory-based tests, such as in vitro assays of blood or urine [128]. While these methods are accurate, they can be time consuming and often require many resources, making them less suited for rapid, point-of-care diagnostics.

Wearable chemical sensors provide a non-invasive method of continuously monitoring various markers in a patient’s body in real-time and still provide valuable information regarding the composition of the individual’s biofluids. Typically, these sensors are composed of a recognition element, which detects the chemical input, and a transduction element, which converts the chemical information into a measurable signal. The exact measurement modality can vary, however, from induced current and voltage changes for electrochemical sensors to colorimeter absorbance or fluorescent intensity changes for optical chemical sensor designs [129]. These non-invasive sensor technologies have the potential to be utilized in a prehospital setting, such as during combat casualty care, to monitor soldiers’ physiological status in real-time, detecting signs of fatigue, dehydration, metabolic stress, or hemorrhagic shock.

#### 5.4.2. Medical Decision Support Applications

Common chemical sensing methods include blood glucose monitoring, lactate measurement, and other biochemical assays performed in controlled laboratory environments. Typically, these sensors monitor an analyte of interest, vitals, or other biomarkers and incorporate those data into a CDS system to deliver accurate, timely, and individualized care. Currently, research is being performed on how these wearable chemical sensors can be used to enhance CDS by enabling remote patient monitoring and providing personalized continuous data for assessing dynamic treatment plans. One CDS example by Winkelmann et al. reported that blood glucose levels in combination with other physiological parameters could be used to identify patients with a high risk of shock [130]. FDA-approved continuous glucose monitoring sensors could be translated into the prehospital care setting to help assess patient priority in austere environments where medical interventions are limited. Another CDS example reported that mortality in posthemorrhagic shock patients was associated with elevated plasma D-lactate levels [131]. This data trend integrated with WHDs for D-lactate measurement, such as recently developed enzyme encapsulated hydrogel networks capable of detecting lactate enantiomers, could serve as a potential tool for improved prehospital triage [132].

Furthermore, incorporating ML/AI into the overall design of chemical sensors can enable them to analyze more complex data streams in real-time and provide detailed insights that can be used to generate CDS tools. Additionally, the integration of ML and AI into these devices will allow for analyses to be performed on-site and eliminate the need for samples to be sent off to a laboratory, which is especially preferred in military settings. Overall, these wearable chemical sensors can significantly reduce wait times for diagnostic test results and enhance the overall efficacy of the medical decision-making process in both civilian and military settings.

#### 5.4.3. State of Development

Current wearable sensors use various transduction mechanisms to detect specific biomarkers in biofluids such as interstitial fluid, saliva, sweat, and tears [128]. Research applications for these sensors are vast, including sweat-monitoring headbands, tear-sensing contact lenses, glucose-measuring mouth guards, and alcohol-detecting temporary tattoos [133,134,135,136]. One main area of interest in the market for wearable chemical sensors is sweat analysis. Wearable sweat sensors come in many forms, including a watch, band, or adhesive patch. These sensors collect sweat from the skin and analyze its composition to provide insight into hydration levels, electrolyte balance, and other biomarkers indicative of a patient’s health. For example, the IDRO sensor functions as a sweat sensor and can provide detailed information regarding sweat biomarkers, as well as both lactate and pH values [137]. Other wearable sweat sensor examples on the market include: the AbsolutSweat real-time sweat tracker for optimal hydration [138], the hDrop biosensor [139], Gx Sweat Patch [140], Epicore Biosystems Discovery Patch^®^ Sweat Collection System [141], and Nix Hydration Biosensor [142]. These sensors gather information regarding both fluid and electrolyte losses and send those data to the user via a smartphone, smartwatch, or computer to alert them regarding their overall hydration levels.

#### 5.4.4. Technology Advantages and Disadvantages

The newly emerging field of wearable chemical sensors offers many capabilities in the management of high-risk trauma patients, however, there are still some limitations. Chemical sensors require a high degree of specificity and sensitivity to properly detect an analyte of interest, and achieving that can be difficult in any type of environment, especially an austere prehospital setting. To avoid complications, sensors need to be designed so they are robust enough to operate in various environmental conditions, including extreme temperatures, humidity, and physical stress. In addition, the size, rigidity, and operational requirements of most chemical sensors do not allow for them to be turned into small, lightweight, wearable devices [143]. The current power sources for these sensors have low energy densities and lead to slow recharging, making them less suited for wearable applications [143]. Lastly, because this technology is so new, the cost of these devices is high, so commercializing them for the public has proven difficult. Overall, the development of easily deployable, wearable chemical sensors for prehospital trauma care has the potential to greatly improve patient outcomes. A summary of chemical WHDs for the prehospital environment and CDS applications is shown in Figure 5.

### 5.5. Electrocardiogram

#### 5.5.1. Overview of Technology

An electrocardiogram (ECG) measures the electrical activity of the heart through its cardiac cycle. Electrical impulses travel with each heartbeat and are measured by electrodes to classify a variety of heart-related issues. Specifically, the time between impulses is measured to identify if a heart has a healthy rhythm, and the strength of the electrical impulse is monitored to identify if parts of the heart are damaged [144]. ECGs are non-invasive and recorded from the surface of the patient’s body [145]. A traditional ECG system consists of up to 12 leads that are split into two separate categories—limb leads and precordial leads. Since the original development of the ECG, there are now several types of ECG monitoring equipment ranging from hardwired continuous monitoring ECGs to wearable ECG technologies [146,147]. Wearable ECG technologies use alternative electrode positions versus traditional ECG systems. These wearables may have multiple leads or a singular lead to gather ECG data [148]. While the 12-lead ECG is currently the gold-standard, wearable technologies of varying lead amounts aim to close the gap.

#### 5.5.2. Medical Decision Support Applications

ECGs are one of the most widely used diagnostic tools in medical decisions regarding cardiac issues. As there are varying levels of abnormal cardiac conditions or cardiac arrhythmias, there is extensive research investigating ECG-driven CDS tools [149]. It is critical for ECG wearables to be as accurate as possible in their CDS role as various cardiac arrhythmias, such as ventricular fibrillation and tachycardia, can trigger cardiac arrest and sudden death [150]. ML models using ECG data from the MIT-BIH and St. Petersburg Institute of Cardiological Technics 12-lead Arrhythmia database were able to classify the five most frequent arrythmia types (left bundle branch blocks (LBBBs), right bundle branch blocks (RBBBs), atrial premature contractions (APCs), premature ventricular contractions (PVCs), paced beat (PB), and fusion beats (FBs)) with high accuracy rates [151]. The use of wearable ECG devices allows for continuous data capture which has been shown to result in a higher rate of atrial fibrillation (AF) medical diagnosis by healthcare professionals [152]. Wearable ECG patches have been used to accurately predict impending hospitalization due to heart failure and predict risk of hospitalization with a sensitivity between 76.0% and 87.5% and a specificity of 85% [153].

#### 5.5.3. State of Development

There are a variety of commercially available ECG WHDs. While there are many different ECG form factors on the market that are either FDA-approved or non-FDA-approved [147], the most common commercial ECG systems are typically in the form of a watch, strap, or an adhesive patch. FDA-approved ECG systems that are in a patch form factor include the ZioPatch (iRhythm Technologies Inc., San Francisco, CA, USA) [154], Wellysis S-Patch (Wellysis Corp., Seoul, Republic of Korea) [155], and the SmartCardia 7L Patch (SmartCardia SA, Lausanne, Switzerland) [156]. In addition to patches, the Polar H10 (Polar Electro Oy, Kempele, Finland) is a wearable, non-FDA-approved strap that incorporates an ECG directly into the strap and is often used for exercise tracking. The Shimmer3 ECG unit (Shimmer Sensing, Dublin, Ireland) makes use of a traditional five-wire, four-lead, non-FDA-approved ECG that is attached to a small form factor module that can be simply attached to a strap that is worn on the body such as on the chest or the wrist. The HeartIN^®^Fit T-shirt (HeartIN INC., Miami, FL, USA) deploys non-FDA-approved ECG sensing technology in clothing. The ECG T-shirt collects data while being worn by a subject and wirelessly sends those data to a transmitter attached to the shirt which then sends the data to the patient’s smartphone [157]. Many of the commercial ECG products are in the form of a watch. The Apple Watch (Apple Inc., Cupertino, CA, USA) is an FDA-approved single-lead wearable that is shown to be able to detect atrial fibrillation [158]. Other commercially available ECG systems in a watch form factor include the Samsung Galaxy Watch (Samsung Electronics Co., Ltd., Suwon-si, South Korea) [159], wrist-worn Fitbit products (Fitbit Inc., San Francisco, CA, USA) [159], wrist-worn Garmin products (Garmin Ltd., Olathe, KS, USA) [160], Google Pixel Watch (Google LLC., Mountain View, CA, USA), and Withings ScanWatch (Withings Inc., Issy-les-Moulineaux, France) [159]. While these devices are “worn”, they require finger contact with the device to record ECG, thus not meeting the WHD definition for this review. Other portable form factors for ECG data capture, which also fall outside the WHD definition, include devices that require holding two electrode lead points with the fingers for a duration [154].

#### 5.5.4. Technology Advantages and Disadvantages

Wearable ECG systems offer a variety of positive capabilities for prehospital care according to experts in the field or related fields. Advantages include faster diagnosis, continuous monitoring, facilitation of screening, and an increase in patient involvement in their health. Negatives of the wearable ECGs according to the same experts were the data processing involved, safety concerns, patient involvement for accuracy, and data overload [161]. Wearable ECGs can have both real-time and non-real-time capabilities, each with their own challenges. Real-time recording of ECGs will require data analysis from the captured data. This can happen in real-time with ML algorithms but requires accurate ML algorithm implementation. Non-real-time recordings of ECGs require the same analysis but it includes the additional step of sending the ECG sensor to those who will analyze the data, whether it be an expert or an ML model [148].

ECG WHDs can benefit the healthcare field and prehospital care with additional improvements. The data being acquired must strike a balance between battery life and memory storage onboard the wearable system without sacrificing accuracy [148]. Real-time analysis and diagnostics are often a desired aspect of wearable devices to inform the medical personnel of patient status. ECG wearables may fail to be adequate in prehospital scenarios as some require data collected in real-time to be analyzed after the fact, resulting in a delayed diagnosis [162]. Real-time detection of arrhythmias using ECGs can be performed but requires additional hardware, software, and sufficient data to be able to begin detecting arrhythmias using the real-time signal [163].

Shortcomings also exist in the placement of the sensors and patient compliance using the ECG wearables correctly to maximize their accuracy. Placement of an ECG is important for gathering data with a quality signal [164]. Watch-based ECG wearables are more susceptible to lower-quality signals due to the variation in how a patient may wear the WHD. Patches are less susceptible to improper placements, and some ECG patches on the market come with a placement template to mitigate this shortcoming [148]. Individuals donning a wearable ECG may also develop skin irritation due to the interface with the patient’s skin [165]. This would in turn affect the adherence and/or placement of the ECG wearable on a patient, further degrading the quality of the data acquired. A summary of ECG WHDs for the prehospital environment and CDS applications is shown in Figure 6.

### 5.6. Seismocardiogram

#### 5.6.1. Overview of Technology

A seismocardiogram (SCG) is a type of microelectromechanical system (MEMS) that detects vibrations of the patient’s chest wall which are induced by heart activity. This approach is a refinement of the ballistocardiogram which aimed to detect heart activity through measuring movement of the whole body [166,167]. As materials science and electronics manufacturing have advanced, the miniaturization of very sensitive yet precise components that form the foundation of MEMS sensors has been made possible. SCG devices typically utilize highly tuned accelerometers in combination with miniature gyroscopes to detect the intensity and direction of subtle vibrations of the chest wall. Alternative sensing modalities have also been explored that include soft sensors functioning as strain gages placed on the skin [168] and millimeter-wave radar reflections [169] to track the same vibrations. The former measures microdeformations in the skin resulting from movement, while the latter measures differences in the signal reflection time to extrapolate movement. Data from these sensors are then analyzed using advanced signal-processing techniques to filter out noise and amplify periodic waveform features that have been associated with the various phases of the cardiac cycle [170]. Further analysis of these processed data features can provide insights into patients’ health by identifying discrete abnormalities that correlate to known illnesses or by monitoring for changes that may indicate a decline in status.

These non-invasive devices may use a strap that wraps around the patient to secure the device to the chest, while others may use an adhesive patch or tape that attaches the device directly to the sternum. Typically, closer, more secure contact with the chest will result in less dampening and a stronger signal, although there may be tradeoffs in the form of increased noise from vibrations caused by other physiological activity like speaking and breathing. Contactless methods, like the millimeter-wave approach, while offering some benefits over those requiring direct contact, may be disproportionately affected by other forms of noise like patient movement. These challenges highlight the importance, and need, for strong computational capabilities to extract meaningful information from the raw SCG signals.

#### 5.6.2. Medical Decision Support Applications

SCG technology is less mature than other physiological sensing modalities, limiting its application for CDS tools to date. However, there are vast opportunities available for the deployment of SCG devices in conjunction with decision support applications. Real-time in-depth knowledge of cardiac performance would enable trauma care providers in the prehospital setting to prioritize treatments that more superficial vital signs might not. Several studies have shown promising results in correlating SCG signals with an array of cardiac conditions, including heart rate, heart rate variability, atrial fibrillation, heart attack, hemorrhage, and ischemia [171,172,173,174]. This could also serve as a means of early detection of worsening cases of non-traumatic chronic illnesses such as heart disease.

#### 5.6.3. State of Development

There has recently been a growing interest in continuous monitoring techniques among caregivers both in and out of hospital settings for the purpose of early detection. There has also been greater interest in personalized health monitoring among the public; as the rise of cardiovascular disease has become more widely known, wearables capable of monitoring heart health long-term have gained enough appeal to foster development that has lain mostly dormant since the invention of the ballistocardiogram. Although there are not yet any commercially available SCG devices, this interest has spurred research efforts aimed at producing one. The MagIC textile-based system uses a vest embedded with two textile electrodes assessing a single ECG lead and a textile strain gauge for tracking respiratory rate which send their signals to a small electronic board equipped with a three-axis accelerometer for measuring vibrations. The device analyzes these signals to estimate an SCG waveform from which diagnostic information can be derived and has been shown to track accurate data outside of a laboratory setting in the presence of the spontaneous movement of daily life [175]. Research efforts have also investigated the development of a contactless method they call RF-SCG which relies upon millimeter-wave radars to sense vibrations [169]. This method utilizes a collection of wave transmitters and receivers along with a novel combination of ML techniques to translate the raw waveforms to an SCG signal. A benefit of using this technology is that it eliminates the need to affix a sensor directly to the chest since the waves can penetrate clothing, although sample data from a contact-based accelerometer are required for the initial training of an algorithm in the pipeline.

#### 5.6.4. Technology Advantages and Disadvantages

Wearable SCG technology is expanding in interest, especially regarding personal health monitoring for at-risk populations as well as those carefully monitoring their personal health. The patch or vest approach is portable and does not affect daily activities, while also allowing for continuous monitoring during the process. The limiting factor for SCG is the combination of an accelerometer and gyroscopic element; selecting an appropriate accelerometer may prove difficult because of their sizes and accuracy. The capability gap currently being addressed is filtering out the noise of the accelerometer measurements where breathing and speaking would cause vibrations. Initial work has shown evidence that this can be fixed, but more work is needed before WHDs can fully use SCG technology. An SCG performs best when the subjects remain still during the measurement process, hence why SCG technology is more prevalent in hospital settings than WHD formats. Regardless of possible drawbacks, SCG wearables coupled with machine learning technology show promise as early predictors for trauma or correlation between other diseases [176]. A summary of SCG WHDs for the prehospital environment and CDS applications is shown in Figure 7.

### 5.7. Temperature

#### 5.7.1. Overview of Technology

Temperature sensors are widely used in everyday life and for a range of human applications such as medical diagnosis and monitoring or exercise physiology. The primary approaches used for tracking temperature changes are through resistance temperature detectors (RTDs) made of different metals with standardized temperature-resistance dynamics or using a temperature-dependent voltage produced from a thermocouple through the Seebeck effect [177]. This temperature change elicits a measurable response, offering valuable insights into what is being studied. Most notably, from a medical perspective, abnormal rises in a person’s body temperature can indicate infection or inflammation, making these sensors potentially indicative of underlying health issues or useful for tracking a person’s daily activities. In the medical field, temperature sensors are widely used due to their compact, portable design and non-invasive measurement approach, allowing for ease of use. Translation of temperature sensors into a wearable format can allow for the ability to continuously measure and transmit data regarding an individual’s health, a step beyond the capabilities of traditional clinical temperature monitors.

#### 5.7.2. Medical Decision Support Applications

Research has shown that the human body undergoes thermal stresses when experiencing a range of diseases, such as fighting off an infection or more chronic situations such as cardiovascular disease, diabetes, cancer, and others [178]. Specifically, an increase in surface and core body temperature has been proven to be a preliminary indicator for serious developing diseases [178]. However, as temperature tracks with so many conditions, there are few clinical decision support tools developed working solely on temperature as an input. One ML model example of extending the monitoring usability of a skin temperature sensor is estimation of core body temperature, a more reliable healthcare metric [179]. However, the performance of a core body temperature estimate is more reliable when paired with other vital signs such as heart rate [180]. Other medical decision support tools utilizing temperature measurement in a suite of physiological inputs include type-2 diabetes management [181], predicting in-hospital cardiac arrest [182], and emergency room triage and hospital admission likelihood [183,184,185].

The use of wearable temperature sensors in CDS applications is less studied, but many of the above examples would be compatible if the WHD accurately tracks temperature. As these sensors can allow for continuous monitoring of patient temperature, this could allow for valuable data needed for early detection of injury or illness to allow for more timely interventions. Further, the incorporation of ML or AI models into these devices can allow for even faster diagnosis and more in-depth analysis of physiological data, aiding in personalized treatment plans and optimizing healthcare.

#### 5.7.3. State of Development

Wearable temperature sensors can offer a non-invasive and convenient method of tracking physiologic thermal patterns. The commercial availability of wearable temperature sensors has grown over the years due to the rising popularity of personalized health monitoring and the growing need for diagnostic tools. Wearable temperature sensors commercially available to the public, such as smartwatches, fitness trackers, and smart clothing, enable individuals to monitor their body temperature and cater to a variety of user preferences and needs. While temperature may not be the sole parameter these devices monitor, it remains an essential measurement for various applications. For example, Tempdrop, a wearable armband sensor, gathers temperature data to provide insight and information into the user’s fertility window [186]. Similarly, a device named CORE utilizes advanced technology to provide real-time core body temperature data for monitoring an individual’s performance. In doing so, CORE can help lower the user’s risk of heat stroke during physical activities [187]. Healthcare technology development continues to focus on wearable sensors capable of tracking temperature as it may provide a preventative medicine functionality with real-time, high sensitivity. As such, both healthcare professionals and direct consumers can utilize these technologies for monitoring body temperature and for tracking general health trends.

#### 5.7.4. Technology Advantages and Disadvantages

In prehospital care settings, wearable temperature sensors offer many advantages including early detection of abnormalities, continuous monitoring, and non-invasive measurement methods. However, these sensors present some challenges in their current state. For example, they can have large variations in accuracy, especially in extreme temperature environments or from patient sweat or movement. They can require constant re-calibration to maintain precision [188]. Additionally, wearable technologies are usually expensive and prone to low battery life [189]. These considerations are crucial when evaluating the suitability of wearable temperature sensors for different applications, especially in the prehospital setting, such as combat casualty care, where time and resources are limited. The incorporation of ML and AI models into these devices may help, but there are still many gaps that need to be addressed before wearable temperature sensors can effectively be used in emergency medical settings. A summary of temperature WHDs for the prehospital environment and CDS applications is shown in Figure 8.

## 6. Discussion

As highlighted in this manuscript, there are a wide range of WHD technologies currently in development that could simplify patient monitoring in the prehospital setting. However, there are some general shortcomings that still need to be addressed. The first is battery life. On the future battlefield where vital sign tracking may be critical for multiple days, it is ideal to be able to run for 24 or more continuous hours or have the capability to easily hot-swap batteries without losing existing vital data. Research is being conducted to harness temperature differences between the air and body or integration of solar or other continuous power sources that will likely be critical in the extended prehospital care setting [190,191,192]. The second shortcoming is in testing and evaluation of these WHDs during evacuation. As the patient is transported in the field by personnel, ambulance, or helicopter, the vibrational noise that will be introduced will need to be assessed for its impact on WHDs, especially for ML-based CDS models built using WHD inputs. This noise, if not accounted for by filtering or through inclusion in training datasets, may result in WHDs being meaningless during this casualty evacuation period of critical importance during prehospital care. The third shortcoming is communication protocols. Most WHDs rely on Bluetooth for wireless communication which allows for tracking multiple sensors for up to 30 m on average [193]. This is probably sufficient for the civilian sector, but, in military medicine, communication security must be considered so that the signal cannot be used by enemy forces. More secure protocols in development will need to be considered in future WHDs designed for use in combat casualty care [194].

In addition, there will be ethical and privacy concerns for WHDs and the development of ML models using data acquired from the WHDs. Data transfer for ML development from WHDs should be considered. WHDs that make use of data transfer using cloud storage have several security and privacy threats [195]. This is minimized if the WHD stores data locally, but this may limit its utility in tracking patient status. ML models developed from the data should also be representative of the general population. Data curated that does not reflect the demographic of the general population can perform poorly on a population that is not accurately reflected in the ML model training dataset. For example, PPG signals inputs to ML models have resulted in worse or inaccurate performance on darker skin tones when the underlying training data do not accurately reflect these different variations of skin tone in the population [196]. As these signals may be used in ML models to derive physiological metrics or make treatment decisions, these ethical considerations are an important aspect to account for when developing ML models with WHDs.

On the CDS side, several additions are needed to better make use of WHDs for making diagnoses, treatment, or triage decisions. The first goal for prehospital decision support is triage for medical evacuation, especially during extended prolonged field care situations where evacuation opportunities may be limited. Oftentimes these decisions cannot be made from one sensor input, often requiring a number of sensors to obtain a full reporting of the subject’s physiological status. This brings with it challenges. First, a central data streaming application will be needed to grab data exported from each sensor type which must have interoperability across a range of sensor options. Second, any triage CDS tool must be capable of identifying what vitals are most critical at certain points in casualty management. Large prehospital datasets with WHDs will be needed to build these triage decision support tools, similar to the MIMIC ICU database [197]. Third, the decision support tool must be flexible to the sensor inputs so that, when one sensor is not available, the triage support tool will make modified decisions instead of halting its use entirely. In a prehospital setting, it cannot be guaranteed that all sensors will be available, so this likely shortcoming needs to be accounted for with CDS tools using WHDs.

Future research directions are still needed to better enable WHDs to support medicine in the prehospital setting. One critical need is developing large, standardized datasets for capturing data streams and comparing against gold standard measurements in a prehospital relevant trauma setting. One such platform could be the lower body negative pressure chamber that allows for collection of simulated hemorrhage datasets to the point of decompensation in human volunteers that has been validated and verified as a standardized hemorrhage platform [198,199,200]. Collecting a wide range of WHD vital signs with this platform will help to confirm what sensors are most applicable to triaging injuries such as hemorrhagic shock. Second, datasets are needed in a standardized use setting to better understand how WHDs will accurately monitor during the motion of a soldier moving, ambulatory transportation, or helicopter evacuation. Datasets to simulate this sort of noise will be critical to better design WHDs and ensure CDS tools constructed can parse through this noise correctly.

## 7. Conclusions

In summary, wearable healthcare devices are of great interest to prehospital medicine, but the prehospital setting has unique challenges that must be considered for WHDs to aid with patient monitoring and triage decisions. Here, we detailed a range of WHD technologies in development with different levels of maturity. However, WHDs alone are insufficient for supporting prehospital medicine without seamless integration with clinical decision support tools to simplify patient status monitoring. CDS strategies for WHDs are in the early research stages but care needs to be taken to ensure they are developed appropriately for prehospital care. Further, more data properly captured with WHDs in relevant trauma environments are needed to properly develop CDS tools capable of managing patient movement and complex trauma scenarios. Pairing WHD technologies with CDS machine learning algorithms could revolutionize prehospital medicine in the future for both civilian and military situations.

## Figures and Tables

**Figure 1 sensors-24-08204-f001:**
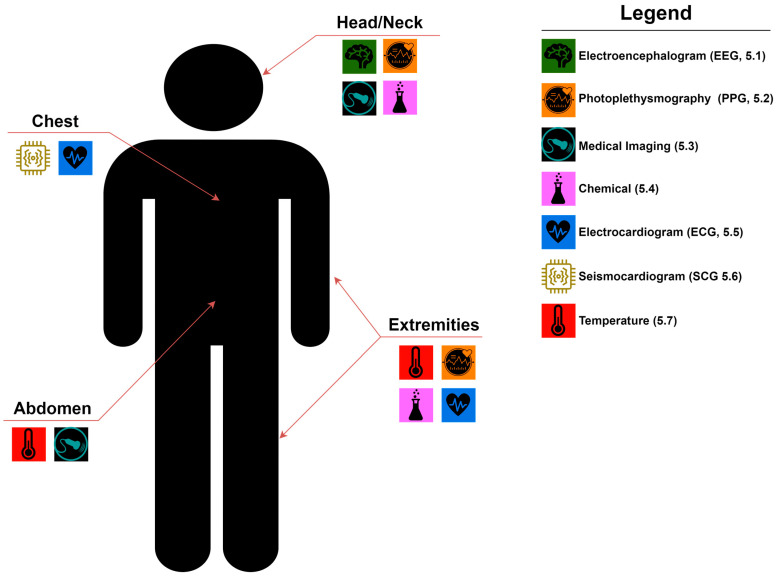
Diagram of Wearable Healthcare Device Placement. For each major body area, relevant WHD signal type is highlighted according to the monitoring capabilities. Article sub-sections for each respective WHD are included in the legend for ease of article navigation.

**Figure 2 sensors-24-08204-f002:**
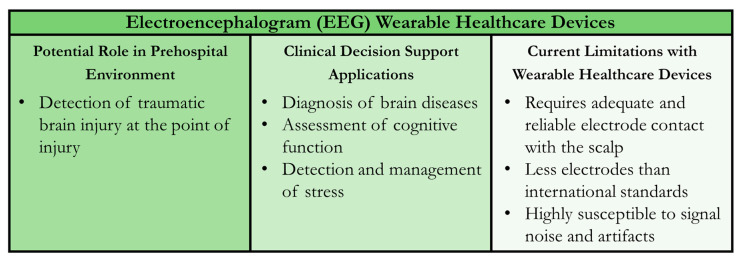
Summary of Wearable Healthcare Devices (WHDs) for Electroencephalogram (EEG) Measurement. Potential uses in the prehospital setting, clinical decision support applications for the sensor technology, and current technology limitations are summarized.

**Figure 3 sensors-24-08204-f003:**
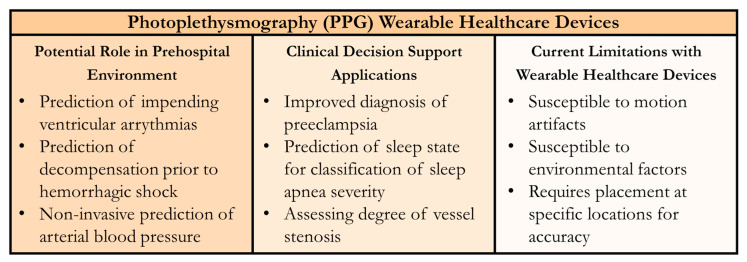
Summary of Wearable Healthcare Devices (WHDs) for Photoplethysmography (PPG) Measurement. Potential uses in the prehospital setting, clinical decision support applications for the sensor technology, and current technology limitations are summarized.

**Figure 4 sensors-24-08204-f004:**
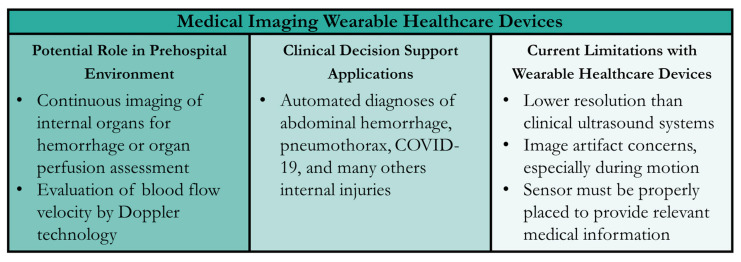
Summary of Wearable Healthcare Devices (WHDs) for Medical Imaging. Potential uses in the prehospital setting, CDS applications for the sensor technology, and current technology limitations are summarized.

**Figure 5 sensors-24-08204-f005:**
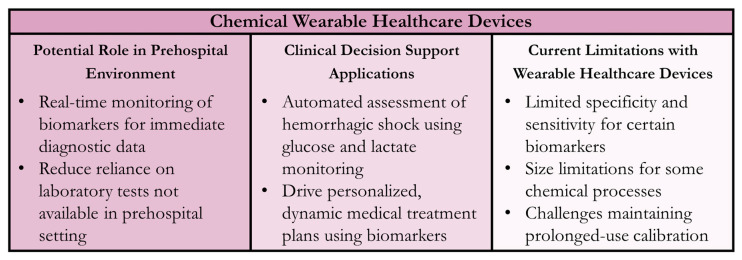
Summary of Wearable Healthcare Devices (WHDs) for Chemical Sensing. Potential uses in the prehospital setting, clinical decision support applications for the sensor technology, and current technology limitations are summarized.

**Figure 6 sensors-24-08204-f006:**
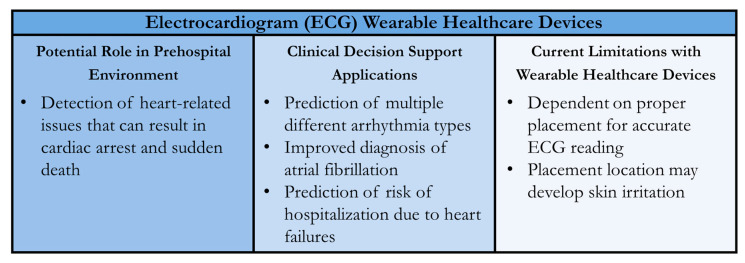
Summary of Wearable Healthcare Devices (WHDs) for Electrocardiogram (ECG) Measurement. Potential uses in the prehospital setting, clinical decision support applications for the sensor technology, and current technology limitations are summarized.

**Figure 7 sensors-24-08204-f007:**
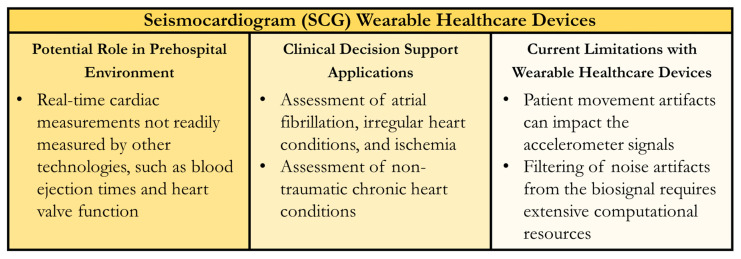
Summary of Wearable Healthcare Devices (WHDs) for Seismocardiogram (SCG) Measurement. Potential uses in the prehospital setting, clinical decision support applications for the sensor technology, and current technology limitations are summarized.

**Figure 8 sensors-24-08204-f008:**
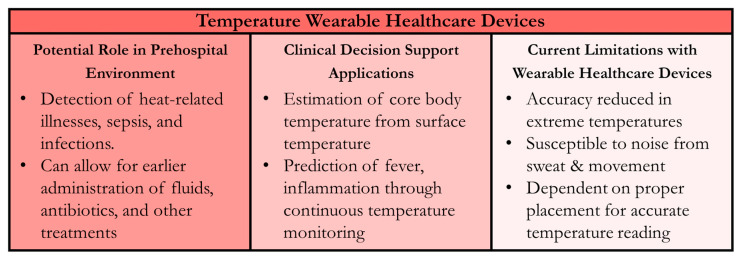
Summary of Wearable Healthcare Devices (WHDs) for Temperature Measurement. Potential uses in the prehospital setting, clinical decision support applications for the sensor technology, and current technology limitations are summarized.

## Data Availability

Data are contained within the article.

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
