# Peer review of "Overview of Wearable Healthcare Devices for Clinical Decision Support in the Prehospital Setting"

_sensors, 2024, doi:10.3390/s24248204_

Round 1

Reviewer 1 Report

Comments and Suggestions for Authors

The paper addresses the challenges encountered in pre-hospital care, particularly within military contexts, and explores how wearable healthcare devices (WHDs) combined with machine learning (ML) can help mitigate these issues. However, further revisions are necessary for the following reasons:

1. Some paragraphs are lengthy and reiterate similar ideas, which may discourage reader engagement. I recommend revising the article to ensure a more concise presentation that captivates the reader's interest.

2. Given the significant implications of WHDs on patient privacy and data security, it would be beneficial to include a brief discussion on ethical considerations, including data privacy, informed consent, and algorithmic bias. 

3. An in-depth and precise discussion of specific ML techniques and their applications to WHD data would be advantageous. For example, a detailed examination of deep learning architectures, such as convolutional neural networks (CNNs) and recurrent neural networks (RNNs), would provide valuable insights.

4. Please consider adding a brief overview of the limitations of current EEG WHDs, such as their sensitivity to noise and limited spatial resolution.

5. Additionally, it would be interesting to explore the potential of wearable ultrasound devices for real-time monitoring of tissue perfusion and organ function.
6. Figures and tables enhance clarity and comprehension. I recommend that the authors consider revising the article to incorporate additional drawings, figures, and tables for better support of the content.

Author Response

The paper addresses the challenges encountered in pre-hospital care, particularly within military contexts, and explores how wearable healthcare devices (WHDs) combined with machine learning (ML) can help mitigate these issues. However, further revisions are necessary for the following reasons:

  1. Some paragraphs are lengthy and reiterate similar ideas, which may discourage reader engagement. I recommend revising the article to ensure a more concise presentation that captivates the reader's interest.

We appreciate the reviewer’s feedback on the article. We have revised the article where possible for clarity and have also added summary graphics into each WHD section to give higher level overviews of the main points in each section. We hope that helps address this comment.

  1. Given the significant implications of WHDs on patient privacy and data security, it would be beneficial to include a brief discussion on ethical considerations, including data privacy, informed consent, and algorithmic bias. 

Thank you for the suggestion. We have added discussion points as suggested to the discussion section.

  1. An in-depth and precise discussion of specific ML techniques and their applications to WHD data would be advantageous. For example, a detailed examination of deep learning architectures, such as convolutional neural networks (CNNs) and recurrent neural networks (RNNs), would provide valuable insights.

Thank you for the suggestion, however, this is outside of the scope of this review article. We have cited review articles for AI models for signal processing in the overview section for reference if additional information on this topic is desired. However, going too much further in depth on this topic we think would make it harder to keep the focus on WHDs.

  1. Please consider adding a brief overview of the limitations of current EEG WHDs, such as their sensitivity to noise and limited spatial resolution.

We have touched more on this limitation with EEG WHDs in the limitation section for that sensor type.

  1. Additionally, it would be interesting to explore the potential of wearable ultrasound devices for real-time monitoring of tissue perfusion and organ function.

Great suggestion, we have highlighted this as another possible application for ultrasound devices for WHDs and the pre-hospital setting in the medical imaging sensor section.

  1. Figures and tables enhance clarity and comprehension. I recommend that the authors consider revising the article to incorporate additional drawings, figures, and tables for better support of the content.

We have taken the reviewer’ s suggestion and added 7 additional figures to better support the content, one figure for each of the 7 WHD sections to better summarize the results of each section. We hope that improves the clarity and comprehension

Reviewer 2 Report

Comments and Suggestions for Authors This paper summarize the current state of wearable sensors, and evaluate its potential utility in pre-hospital setting, with specific focus in military setting.    This paper is well structured, backed up by a rich amount of information, and reads more like a white paper or market research report rather than a research paper. Assuming the standard of a review paper, here are a few comments:   1) the paper should have a dedicated methods section providing details on search strategy, inclusions/exclusion criteria   2) the paper also need to refer to existing reviews that share some similarity of the paper, and articulate the contribution of this paper compared with other review papers   3) although the paper center-around clinical decision support, it is not clear what kind of decisions are relevant pre-hospital settings, and they are different or similar from those in other settings such as critical cares    4) the paper emphasize the critical role of trends (e.g., line 109). It is not clear the authors refer to short term trends or long term trend, as it can impact for how long those device needs to be weared by the patients to collect sufficient data for reliable trends    5) Many of the devices introduced are commercial or research-in-progress devices. If possible, please provide evidence/citations for those devices, either validating their measurement or studies that use those devices in real-world settings. Comparison of those devices in a table format will make the information more digestable. 

Author Response

This paper summarize the current state of wearable sensors, and evaluate its potential utility in pre-hospital setting, with specific focus in military setting.    This paper is well structured, backed up by a rich amount of information, and reads more like a white paper or market research report rather than a research paper. Assuming the standard of a review paper, here are a few comments:  

1) the paper should have a dedicated methods section providing details on search strategy, inclusions/exclusion criteria  

We appreciate the reviewer taking the time to review our manuscript submission. We have added a methods section that better describes the approach taken to identify the WHDs and CDS information

2) the paper also need to refer to existing reviews that share some similarity of the paper, and articulate the contribution of this paper compared with other review papers  

We have added reference to other review articles on similar topics to this one and explained why this article and its focus on the prehospital setting and CDS applications is different from others. This was added in the introduction section

3) although the paper center-around clinical decision support, it is not clear what kind of decisions are relevant pre-hospital settings, and they are different or similar from those in other settings such as critical cares   

We have tried to clarify what CDS applications are most pertinent to the prehospital setting in section 3. There are a lot of overlap to critical care but are separated mainly by the small footprint available prehospital and less skilled medical personnel using the information, requiring more robust CDS tools to aid in this setting

4) the paper emphasize the critical role of trends (e.g., line 109). It is not clear the authors refer to short term trends or long term trend, as it can impact for how long those device needs to be weared by the patients to collect sufficient data for reliable trends

Thank you for the feedback, we have added additional impact to inform of the potential length of time to establish trends.

5) Many of the devices introduced are commercial or research-in-progress devices. If possible, please provide evidence/citations for those devices, either validating their measurement or studies that use those devices in real-world settings. Comparison of those devices in a table format will make the information more digestable. 

We have added summary graphics after each of the 7 WHD sections to aid with information digest. References throughout these sections refer to validation studies with some of the WHDs if they have matured to that point

Reviewer 3 Report

Comments and Suggestions for Authors

This work is a comprehensive review of the state of the art on wearable health care devices summarizing the challenges of the wearable sensor technology to be suitable to support pre-hospital medicine. A lot of valuable information is given in a well-structured manuscript. As the text is very dense with a lot of medical terms, I suggest to the authors, in order to make it more readable, to include some tables summarizing the key elements/results presented in each sub-section of section 4 of the manuscript.

Author Response

This work is a comprehensive review of the state of the art on wearable health care devices summarizing the challenges of the wearable sensor technology to be suitable to support pre-hospital medicine. A lot of valuable information is given in a well-structured manuscript.

As the text is very dense with a lot of medical terms, I suggest to the authors, in order to make it more readable, to include some tables summarizing the key elements/results presented in each sub-section of section 4 of the manuscript.

We appreciate the reviewer taking the time to review our manuscript. We have taken the reviewer’ s suggestion and added 7 additional figures to better support the content, one figure for each of the 7 WHD sections to better summarize the results of each section. We hope that improves the readability of the manuscript

Round 2

Reviewer 1 Report

Comments and Suggestions for Authors

Accept